# Identification of Aquifer and Pumped Well Parameters Using the Data Hidden in Non-Linear Losses

**Kosta Urumović** [1], **Josip Terzić** [1], **Jasna Kopić** [2,*] **and Ivan Kosović** [1]

1    Croatian Geological Survey, Ul. Milana Sachsa 2, 10000 Zagreb, Croatia; kurumovic@hgi-cgs.hr (K.U.); jterzic@hgi-cgs.hr (J.T.); ikosovic@hgi-cgs.hr (I.K.)
2    Vinkovci Water and Wastewater Association Ltd., Ul. Dragutina Žanića Karle 47a, 32100 Vinkovci, Croatia
*    Correspondence: jasna.kopic@vvk.hr

**Abstract:** During the pumping of wells, the groundwater level drawdown, as measured in the pumped well, is increased by non-linear losses caused by the water flow velocity through the well screens. This undermines the adequacy of the direct use of the measured drawdown data in the well for the purpose of the realistic identification of the effective well radius and aquifer parameters. This anomaly is avoided by reshaping the drawdown function into a function of the specific drawdown $s_w/Q$ of the pumped well. This reshaping simplifies the exclusion of non-linear losses from the sequence of measured data of the water level in the well at the position of the effective radius of the pumped well. Combining the data of linear losses and the respective pumping rate of the pumped well, a function of the specific drawdown of the radial flow $s_w/Q$ was formed. This function describes the aquifer parameter relations during the respective test pumping. A consistent sequence of the function of the specific drawdown $s_w/Q$ of the pumped well reveals the actual value of the coefficient of nonlinear losses. Moreover, the specific drawdown function enables the reliable estimation of aquifer transmissivity using only the pumped well drawdown data.

**Keywords:** well-loss parameters; specific drawdown; effective well radius; well-loss constant; transmissivity

## 1. Introduction

The step-drawdown test is a standard and widely used method of defining the pumping capacity of a pumped well. Monitoring the oscillations of the water level in the well and, if possible, in the nearby observation borehole, along with the constant measurement of the pumping rate, enables the determination of the actual capacity of the well, the recommended pumping regime, and the aquifer and well parameters' determination.

The data on well pumping capacity and resulting groundwater level are the most valuable information for identifying aquifer parameters and the condition of the pumped well. The measured drawdown in the well is the sum of the head losses of individual components [1], e.g., linear head losses are amplified by the non-linear losses (caused by the velocity of groundwater flow from the aquifer into the well construction) in the pumped well. Such circumstances generate the paradox of unknown theoretical drawdown of the radial flow in the pumped well. The theoretical drawdown of linear flow losses in the pumped well, $s_w$, is at the position of the effective well radius, $r_w$ [2,3], i.e., at the position of the hypothetical inner border of the radial flow function. It is positioned above the measured water drop during the pumping of the well.

As opposed to linear losses that depend on the discharge and aquifer parameters, non-linear losses depend on local flow velocity and well loss constant $C_w$, which causes the turbulence in flow [2,4–8].

The main contribution of this research in hydrogeology is a novel approach to step drawdown tests' interpretation, resulting in a substantial increase in the precision of the non-linear well loss coefficient determination. In addition, the emphasis on the utilization

of the pumped well as a main observation object for the purpose of aquifer properties determination enables the simpler and more cost-efficient identification of hydrological conditions in hydrogeological research.

## 2. Description of the Test Fields and Analysis Methodology

### 2.1. General Information on the Analyzed Test Fields

Analyses of step-drawdown test data and the identification of aquifer parameters and tested wells are presented for four test fields in northern parts of the Republic of Croatia (Figure 1). The general features of wells in these fields are summarized in Table 1.

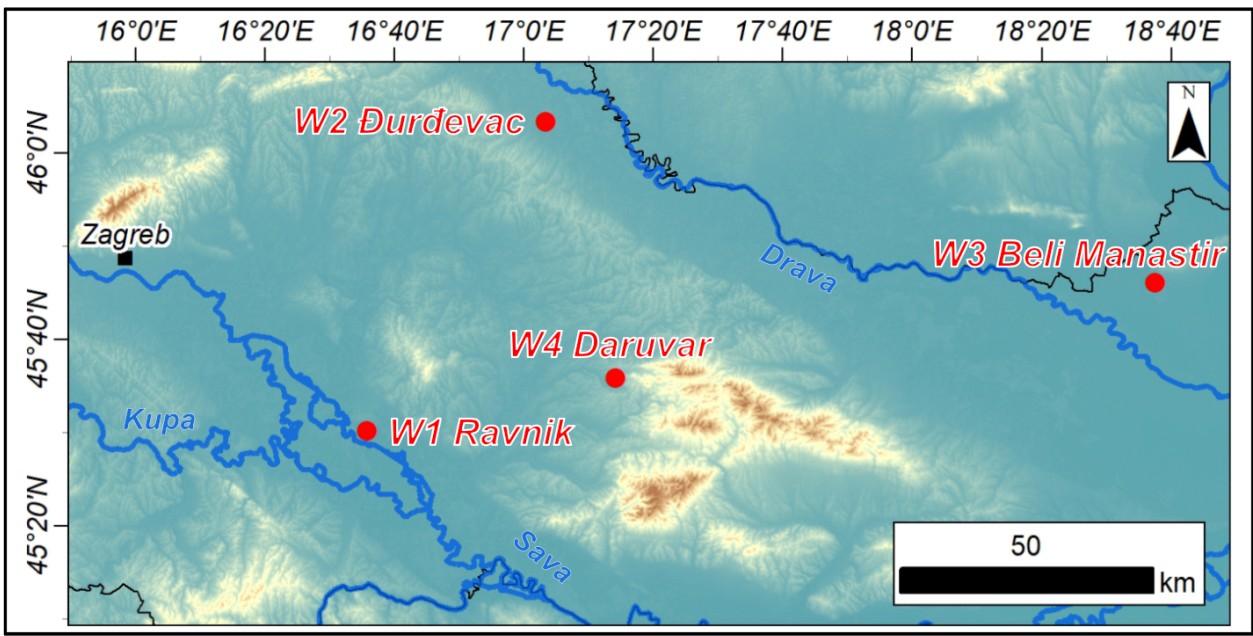

**Figure 1.** Map of the investigated area.

**Table 1.** General characteristics of four test fields in Croatia (Figure 1) used as examples of the application of specific drawdown in the identification of well parameters and the affected aquifer.

| Test Field and Geological Region | Aquifer Type | Water-Bearing Beds | Screens Interval (m)–(m) | Aquifer Thickness (m) | Initial Head Depth (m) | Max. Test Drawdown (m) | Screen Pipe Diameter (m) |
|---|---|---|---|---|---|---|---|
| W1 Ravnik—Sava terrace | Confined sandy aquifer | Uniform medium-grained sand | 82–112 | 38 | 6.75 | 6.81 | 0.4063 |
| W2 Đurđevac—Drava plain | Unconfined gravelly aquifer | Gravel and sand mixtures | 9–22 27–41 49–52 57–62 | 55 | 3.06 | 2.55 | 0.4063 |
| W3 Livade—Baranja terrace | Confined sandy aquifer | Uniform medium-grained sand | 22–26 29–37 | 14 | 10.18 | 13.58 | 0.4063 |
| W4 Daruvar spa—fractured aquifer of the western part of Mt. Papuk | Confined karst aquifer (deeper thermal aquifer) | Fractures and caverns with calcite | 128–134 140–187 | 53 | 5.35 | 0.59 | 0.139/ 0.125 |

The first two examples are from the data archive of step-tests conducted during the building of the respective wells in 2010 and 2013. An exemplary test field was W1-Ravnik, where the pumped well captured a confined aquifer of sandy deposits in a broad valley of the Sava River. The second experimental field, W2-Đurđevac, is located in the vast valley of

the river Drava. The well captures an approximately 65 m thick unconfined heterogeneous gravel aquifer. The length of the installed screens is 35 m. The upper screen of well W2 (Table 1) was installed 8 m below the stationary groundwater level, so the flow also had the characteristics of partial penetration. A step-test was performed on both wells, maintaining three steps of constant yield. Testing lasted 12 and 18 h, respectively, and the first step lasted 6 h in both cases.

Well W-3 is an example of a step-test conducted in 2021, after the rehabilitation of the well, which was carried out with frequent and coordinated measurements of the drawdown of the water level and the respective pumping rate of the well, enabling the spontaneous continuity of the specific drawdown. The well captured a shallow sandy aquifer (Table 1), located in the terraced area of the Danube Plain in Baranja (Figure 1). The well was built about thirty years ago but has not been used. The testing was carried out in three steps, it lasted a total of 4 h, and the first pumping step took 80 min.

The fourth example of the step test was well W4, located inside the existing thermal spring of Daruvar Spa. The well captures thermal water from the fractured and cavernous aquifer of Triassic dolomite and breccia. The screens were installed at intervals between 128 and 134 and between 140 and 187 m deep. The well has not been used for a long time, and the pumping test was performed to determine the well capacity within the scientific project Hythec, funded by the Croatian Science Foundation.

### 2.2. Methodology—Identification of Parameters Using the Function of Specific Drawdown

The analytical procedure for identifying the parameters of the aquifer and the pumped well through the specific drawdown function was primarily formulated following Jacob's approximation of the transient radial laminar flow [1,2,9,10]. The drawdown measured in the pumped well $s_{mw}$ included linear losses of the well, $s_w = B_r Q$, and non-linear losses, $s_n = CQ^2$ [2], as a section of the total losses. These were the result of a continuous function of groundwater level losses presented by the linear equation of the radial flow through the aquifer and part of the well casing up to the lowest level at the hypothetical position of effective well radius $r_w$:

$$s_{m,w} = s_w + s_n = B_{r_w} Q + CQ^2 \tag{1}$$

The ratio was primarily proposed for steady-state conditions. The exact ratio can be applied by analyzing the transient development of the specific drawdown at the location of the pumped well at the position of the effective radius $r_w$ or as a successive sequence of quasi-steady positions between the pumped well and nearby piezometers. In any case, the non-linear well losses, $s_n = C_w Q^2$, (caused by the turbulence in the u well construction) depend on the characteristics of the well construction and of the flow rate of the well and represent a simple section of the total losses of the pumped well. When analyzing the transient development of specific drawdown, Equation (1) is transformed into the equation of the specific drawdown as a function of the effect of the aquifer parameters during the test time, independent of the capacity of the well [11]:

$$\frac{s_{mw}}{Q} - C_w Q = \left( \frac{s_{r_w}}{Q} \right)_t = B_{f_{rw},t} \left[ TL^{-2} \right] \tag{2}$$

The validity of the set relationship is realized only if the constant $C = C_w$, i.e., the actual value of the non-linear well loss constant of the respective well. The exact $C_w$ value in Equation (2) is demonstrated by the continuity of the development of a specific drawdown before and after each rapid change in the well pumping rate. The real value of the coefficient $C_w$ achieves the uniqueness of the overall sequence of the specific drawdown curve, everywhere before and after short-term disturbances caused by a rapid change in the pumping rate and the consequent transition of the amount of specific drawdown $s_w/Q$ in the previous step to the same ratio $s_w/Q$ in the next step, but with different values of $s$ and $Q$, and for each subsequent testing step. Such a curve of specific drawdown represents

the continuity of the radial sequence of linear losses through the aquifer, and the maximum value of these linear losses is at the position of the fictitious effective radius of the well $r_w$.

With the graphic construction of the semi-log diagram of the specific drawdown of the well, the effect of the coefficient of nonlinear losses $C_w[T^2L^{-5}]$ can be easily identified and its size can be verified. In the step-drawdown test, a rapid change in pumping rate causes a rapid change in the drawdown, demonstrating the inertia of the response of the aquifer system at the transition of the specific drawdown in Equation (2) from the value of $s_w/Q$ during the previous step to the same specific drawdown, but with different amounts of $s_w$ and $Q$ The effect of the deflection during the transition is measured by the value of the included constant $C_w$. When overestimated ($C+ > C_w$), the value of the coefficient of nonlinear losses of the well is included in Equation (2); after a rapid change in pumping rate, the $s/Q$ curve descends, expressing the sequence of ratios of specific drawdown with smaller amounts of $s/Q$, and if it is underestimated, as in $C- < C_w$, the curve of the following of steps will form a new sequence above the previous one. Examples of curve development $(s_w/Q_t)$ in chapter 3 llustrate examples of the specific drawdown curve at the real value of the coefficient $C_w$, which is traced along a corridor bounded by the curves of the same equation, but with slightly overestimated C+ and underestimated C-coefficients of nonlinear losses. By identifying non-linear losses for the data of individual measured drawdown, $s_{m.t}$, a file of specific linear flow drawdown $(s_{w,t}/Q_t)$ is formed and these data are reference values for the radial flow equations for the position of the fictitious effective radius of the well, $r_{w,t}$.

The specific drawdown $s_{w,t}/Q_t$ $[T^1L^{-2}]$ in the confined aquifer (Equation (2)) is expressed using the transient radial flow drawdown equation in Cooper and Jacob's [10] method using unified variables, the well drawdown $s_{w,t}[L]$ and its discharge $Q_t$ $[L^3T^{-1}]$, in a unique variable of aquifer properties and pumping duration $B_{r_w,t}[TL^{-2}]$ in Equation (2):

$$\frac{s_{mw,t_n}}{Q_{t_n}} - C_w Q_{t_n} = \frac{s_{w,t_n}}{Q_{t_n}} = \frac{2.3}{4\pi T} log_{10} \frac{2.25 T t_n}{r_w^2 S} \tag{3}$$

The specific drawdown function (Equation (3)) is independent of the discharge of the well. The basic sequence of the curve $s/Q = f(t)$ maintains the continuity appropriate to the properties of the aquifer. The transmissivity of the aquifer is shown below [1]:

$$T = \frac{2.3}{4\pi\Delta(s_w/Q)} log_{10}(t_2/t_1) = \frac{0.183}{\Delta(s_w/Q)} \tag{4}$$

It is inversely proportional to the increment $\Delta(s/Q) = (s_w/Q)_2 - (s_w/Q)_1$ for the range $t_2 = 10t_1$, $log10(t_2/t_1) = 1$ in Equation (4). A unique straight-line sequence $s/Q$ can be unambiguously determined if the $s_t$ and $Q_t$ measurements are coordinated, so the same straight-line sequence from the first step is confirmed through each subsequent step-drawdown test step. The left extension of the same line intersects the zero value $s/Q = 0$ at $t_0$, and is the same as for $s = 0$ in Jacob's method:

$$\frac{2.25 T t_o}{r_w^2 S} = 1 \tag{5}$$

The alternative procedure is the calculation of the effective well diameter $r_w$, according to Equation (3) for relevant values $(s_w/Q)_t$ at individual moments $t_n$:

$$r_w = \sqrt{\frac{2.25 T_w t_n}{S * EXP(4\pi(s_w/Q)_{t_n} T_{w,P})}} \tag{6}$$

Linkage between $S$ and $r_w$ can be avoided using a satellite piezometer, and in the case of a single well test, using literature data on water and aquifer compressibility is a common procedure.

The exact value of the effective radius of the well can also be analyzed using the formulation of the specific drawdown for a successive series of pseudo-steady states between $s_{w,t}/Q_t$ and the satellite piezometer $(s_P/Q)_t$ by applying Thiem's [12] formulation of the radial flow:

$$r_w = \frac{r_p}{EXP\left[2\pi T\left((s_w/Q)_{t_n} - (s_P/Q)_{t_n}\right)\right]}, \tag{7}$$

thus objectivizing the calculations of effective radius of the pumped well and applying the relevant data for the calculation of the storage coefficient by using transient methods. The same pseudo-steady model can be used for the calculation of the local transmissivity coefficient of the aquifer:

$$T_{w-p} = \frac{ln(r_p/r_w)}{2\pi((s_w/Q)_{t_n} - (s_p/Q)_{t_n})} \tag{8}$$

In this case, the effective radius of the wells $r_w$ from the transient solutions should be used.

The testing of the well of unconfined aquifer W2 was followed by different processes of internal stress and radial drainage of the aquifer. These took place at different time intervals. Immediately after the start of pumping, the decompression of the water and the compaction of water-bearing formations occurred. This was expressed in the first few minutes. In this phase, the development of the drawdown is controlled according to elastic storativity and is equivalent to the behavior in a confined aquifer. The transitional phase begins with the appearance of the delayed yield, which slows down the drawdown to a minimum and then gradually increases and continues in the third phase as a time-drawdown curve governed by the specific yield constant (an indicator of aquifer drainage) [13]. When testing well W2, only the end of the transition phase was registered. In the fourth minute, the value of drawdown was already connected to the bi-logarithmic sequence of Jacob's approximation of the Theis (1935) [14] equation of drawdown, $s = f(t)$, applicable for the analysis and identification of transmissivity, T, and specific yield, Sy, of the aquifer [15].

The analysis models of confined aquifers can be used for analogous relations of unconfined aquifers as well as in cases of the partial penetration of the aquifer, in case the maximum drawdown of less than a quarter of the saturated thickness of the aquifer $h_0$ ($s/h_0 < 0.25$) is achieved. In such cases, it is necessary to use the corrected drawdown $s_c$ [9,16] (Jacob, 1944, 1963):

$$s_c = s - s^2/2h_0 \tag{9}$$

in which, during partial penetration [5], the thickness of the aquifer $h_0$ is replaced by the thickness of the active penetration of the well $l$ and the transmissivity of the aquifer is replaced by $T = Kh_0$. The set conditions are more demanding than the USDI [17] standard. The unconfined aquifer also changes during the drainage of the saturated thickness; $h = h_0 - s$. The storage coefficient also changes with the duration of pumping. At the beginning of pumping, the effect of the elastic properties of the aquifer dominates and the storage is approximately equivalent to that of an unconfined aquifer. The appearance of gravitational drainage, delayed yield, gradually becomes recognizable after ten minutes, and the real identification of the specific yield coefficient is only possible after long-term pumping, in some cases even of monthly dimensions [18].

Steady radial flow in an unconfined aquifer is expressed by the Dupuit–Forchheimer [19,20] equation arranged in the form of Thiem's [12] radial flow equation. The difference in the square of the thickness of the unconfined aquifer between the piezometer $h_p$ and the well $h_w$, $(h_p{}^2 - h_w{}^2)$ in the numerator of the Thiem–Dupuit equation is rearranged according to the binomial sequence:

$$Q = \pi K \frac{\left(h_p^2 - h_w^2\right)}{ln(r_p/r_w)} = \pi K \frac{(h_p - h_w)(h_p + h_w)}{ln(r_p/r_w)} \tag{10}$$

Expressing the groundwater level heights in the denominator of Equation (10) by drawdown below the initial height $h_0$ in the well $h_w = (h_0 - s_w)$ and the piezometer $h_p = (h_0 - s_p)$ provides the relative thickness of the aquifer, the difference in drawdowns $(s_w - s_p)$ and their effects:

$$Q = \frac{\pi K_{w-p}(2h_0 - s_w - s_p)(s_w - s_p)}{ln(r_p/r_w)} \tag{11}$$

When the partial penetration length is $l$, the hydraulic conductivity expressed by the difference in specific drawdown between the pumped well and the neighboring piezometer is obtained from Equation (11):

$$K_{w-p} = \frac{ln(r_p/r_w)}{\pi(2l - s_w - s_p)((s_w/Q) - (s_p/Q))} \tag{12}$$

And the effective well diameter is:

$$r_w = \frac{r_p}{EXP[\pi K(2l - s_w - s_p)((s_w/Q) - (s_p/Q))]} \tag{13}$$

The term $(2h_0 - s_w - s_p) = (2l - s_w - s_p) \equiv (h_w + h_p)$ in the denominator of Equations (12) and (13) is the sum of the thickness of saturated deposits [21] at the locations of the effective radius of the well $r_w$ and the piezometer $r_p$, so the mean thickness of the aquifer between the radius $r_w$ and $r_p$ can be approximately expressed as $(2h_0 - s_w - s_p)/2$, and its hydraulic conductivity $K(2h_0 - s_w - s_p) \approx 2T_{w-p}$ [20]. The difference in specific drawdown between the pumped well $(s_w/Q)$ and the satellite piezometer $(s_p/Q)$ summarizes the effect of external variables on the analyzed area of the radial flow. The internal limit of the radial flow is the "effective well radius", (Equation (7), and the basic parameter of the flow gradient is the transmissivity $T_{w-p}$ of the sediment (Equation (8)) between the well and the piezometer.

The transmissivity of deposits of initial saturated thickness at the same location is:

$$T_{h_o} = \frac{2h_o}{2h_o - s_w - s_p}T_{w-p} \tag{14}$$

And the mean hydraulic conductivity is:

$$K = \frac{2T_{w-p}}{2h_o - s_w - s_p}. \tag{15}$$

As presented in this chapter, the specific drawdown function introduces derives several options for determining the well and aquifer properties, simply by rearranging the well-known drawdown equation (Equation (1)), and surpasses the presence of neighboring observation piezometers in some situations.

## 3. Results—Identification of Test Fields Parameters Using the Specific Drawdown Function

Step-test analyses for four test fields (Figure 1, Table 1) were presented graphically using bi-log diagrams $(s_w/Q)_t = f(log10t)$. The first two fields, W1 and W2, were explored ten years ago. In both fields, satellite piezometers with screens placed in the same aquifer layers as the respective wells were also observed. A confined sandy aquifer was tested on the first well, and an unconfined gravel aquifer with thin interlayers of clay sediments was tested on the second well. The third field was an old, unused well of a confined sandy aquifer. A short-term step test of that lonely well was carried out after its revitalization. Interpretations of the testing of all three wells were illustrated graphically on semi-log plots. The sequence of the magnitude of the specific drawdown in the linear losses of the wells during the step-test is presented as a function of the logarithm of the time elapsed since the

start of pumping. During the testing of the first two test fields, the groundwater levels in the well were continuously recorded with loggers during the entire testing phase, and the projected constant pumping rate of the well in individual pumping steps was corrected periodically. The third and fourth step-test examples, W3 and W4, were programmed and executed with the concept of spontaneous specific drawdown as a function of aquifer properties during testing. Additionally, W3 and W4 test fields did not have an observation piezometer. Therefore, only the results of specific drawdown analyses and transmissivity results are presented. The raw measuring data were presented in Supplementary Material in form of Excel file Supplementary_Data.xlsx with sheets W1, W2, W3 and W4, respectively.

*3.1. W1 Ravnik*

Well W1, in Ravnik (Table 1, Figure 2), captures a sandy, confined aquifer. The step-drawdown test was carried out in three steps (Figure 2) for a total of 12 h, and the first step lasted 6 h. The interpretation of the step-test is presented as a general case of identifying the parameters of a closed aquifer by applying the specific drawdown function, $s/Q = f(log10t)$. From the archived recordings of the water level during the testing, the drawdown data were selected with the frequency at which the moments of measurement and the continuity of the development of the specific drawdown in the bi-log scale were visible (Figure 2). For the measured drawdowns, $s_m$, in the selected measurement times, the pumping rate of the well $Q$ was estimated and a list was formed, suitable for the graphic presentation of the operational data file $s_w/Q = ((s_m/Q) - CQ)$ and the specific drawdown of linear losses (Equation (2)). Curves of linear losses $s/Q$ were plotted for three values: underestimated ($C-$), real ($Cw$) and overestimated ($C+$) coefficients of nonlinear losses. In the first step of the step-drawdown test, the data on the semi-log diagram $s_w/Q = f(log10t)$ already formed a linear sequence (Figure 2) equivalent to the Cooper–Jacob diagram of the development of drawdown as a function of time, but with a different meaning. The diagram $s_w/Q = f(log10t)$ is independent of the pumping rate of the well and represents a real function of the properties of the affected aquifer. Gradual slight changes in the pumping rate spontaneously changed the drawdown maintaining the specific drawdown only as a function of aquifer properties. A more significant rapid change in the yield of the well caused a deformation of the $s_w/Q = f(log(t))$ curve caused by the inertia of the aquifer system when moving from the direction $s_w/Q = f(log10t)$, formed during the previous pumping step, to its extension with the same ratio $s/Q$ at the next step of the step-test (Figure 2), but with different values of $Q$ and relative to the appropriate effect of the properties of the aquifer. Thus, the shape of the line before and after the induced discontinuity in the curve $s_w/Q = f(log10t)$ identically expresses the transmissivity of the aquifer. Such curve shaping is a graphical difference in relation to Cooper and Jacob's graphical method.

A rapid increase in the pumping rate caused a marked increase in drawdown, and then the reverse sequence occurred because the excessive specific drawdown returned to the real value of specific drawdown appropriate to the tested aquifer. There was an asymptotic return of the curve to the extension of the previous direction, governed by the relations of the parameters expressed in Equation (3). This sequence of the curve of the specific drawdown of the well W1 (Figure 2) could be achieved if the coefficient of nonlinear losses in Equation (2), $Cw = 4.7 * 10^{-8} \left( day^2/m^5 \right)$, was realistic. The verification of the accuracy of $Cw$ and the realistic slope of the line traced by the Equation (4) was simply confirmed by the opposite effects of overestimated $C+ = 8 * 10^{-8} \left( day^2/m^{-5} \right)$ and underestimated value $C- = 1 * 10^{-8} \left( day^2/m^5 \right)$ (Figure 2). The joint graphic presentation of the results with slightly higher and slightly lower $C$ created a corridor through which the direction $s_{w,t}/Q_t = f(log10(t))$ could pass, verifying the respective result. The slope of the realized line $s_w/Q = f(log(t))$ was proportional to the transmissivity coefficient of the aquifer (Equation (4)). The intersection of that line with zero value, $s_w/Q = 0$, shows the

time $t_0$ from Equation (5) for the calculation of the effective well radius $r_w$ or the storage coefficient $S$.

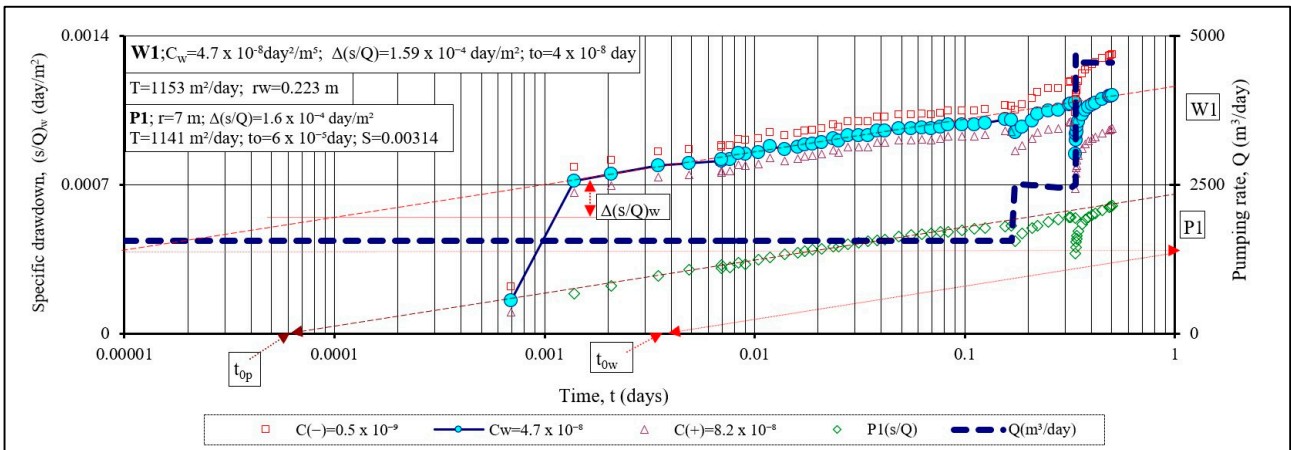

**Figure 2.** Specific drawdown s/Q during the step-drawdown test of well W1 at realistic C = $4.7 \times 10^{-8}$ coefficient of nonlinear losses in the corridor of overestimated C(+) and underestimated C(−) (Equation (4)).

An equivalent analysis of aquifer parameters using the Cooper–Jacob method in Figure 3 offers a comparison of the results using the methods of drawdown and specific drawdown. All parameters identified by the drawdown method were slightly below the respective parameters according to the specific drawdown method. The biggest discrepancy was in the transmissivity $T$ value, which was about 6% higher when analyzed by the specific drawdown method as a consequence of eliminating the effect of non-linear losses. By forming a list of specific drawdowns in the linear flow at the position of the effective radius $r_w$, the conditions for realistic observation of the development of a successive series of spatial positions of the cone of depression between the satellite piezometer P1 and the pumped well W1 were realized using the pseudo-steady states procedure (Equations (7) and (8)).

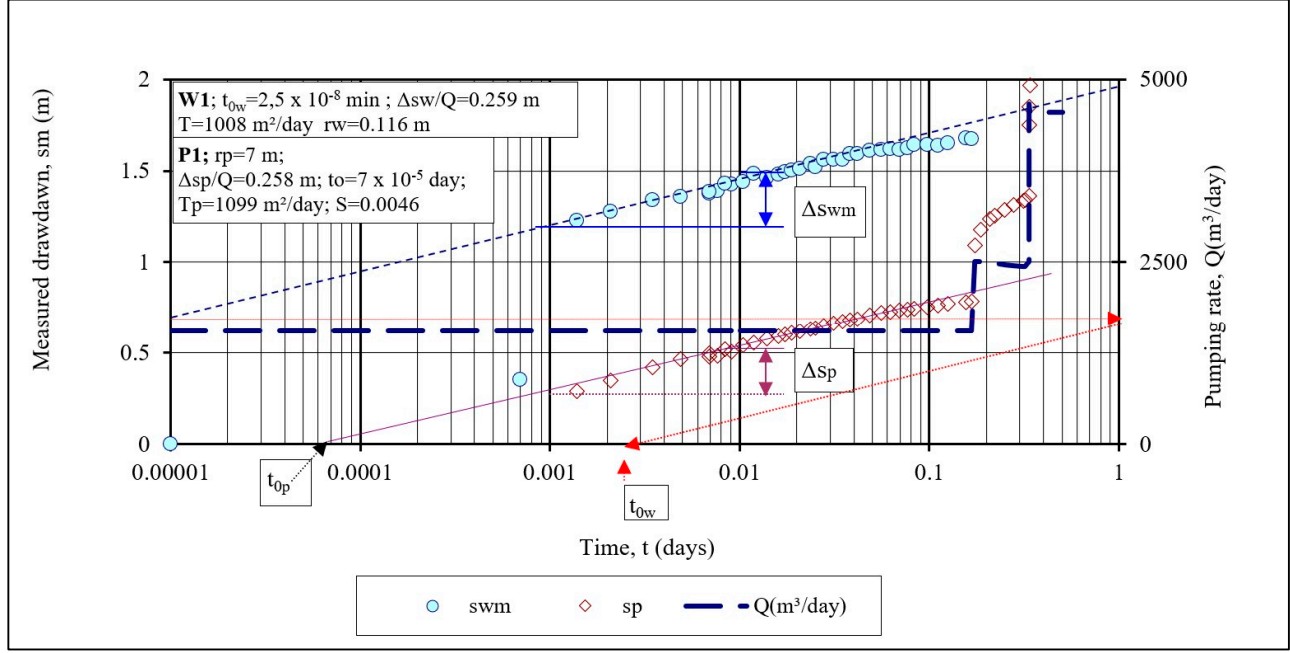

**Figure 3.** Graphical representation of the results of analysis of the measured drawdown (cm) of the water level in the pumped well W1 and piezometer P-1 during the first step of the step-drawdown test [10].

Calculations of the effective radius of the well W1 are shown in Figure 4. Using the respective equations, Figure 5 shows the transmissivity of the aquifer by following a series of quasi-steady positions during pumping test of the well. The operational data file of the specific drawdown of the linear losses of the well, $s_w/Q = ((s_m/Q) - CQ)$ (Equation (2)), and the observation piezometer $s_p/Q$ for simultaneous measurements, offer a reliable presentation of the sequence of quasi-steady positions during the pumping test of the well W1 for calculating the transmissivity of the aquifer (Equation (8), Figure 4) and the effective radius of the well (Equation (7), Figure 4). When calculating with these equations of steady flow, calculations of their values are available with equations for transient flow in which the effects of well losses are excluded.

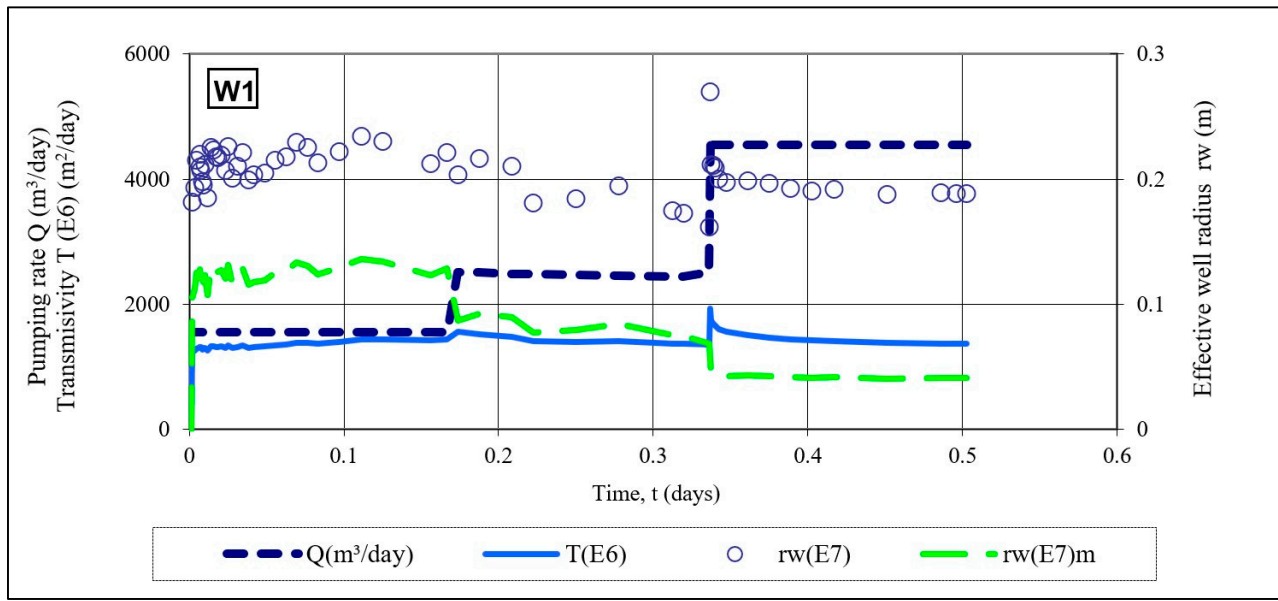

**Figure 4.** The coefficient of transmissivity and the effective radius of the well $r_w$ identified using the specific drawdown function and $r_{w,m}$ according to the measured data using the Cooper–Jacob method. Label E5—the corresponding equation.

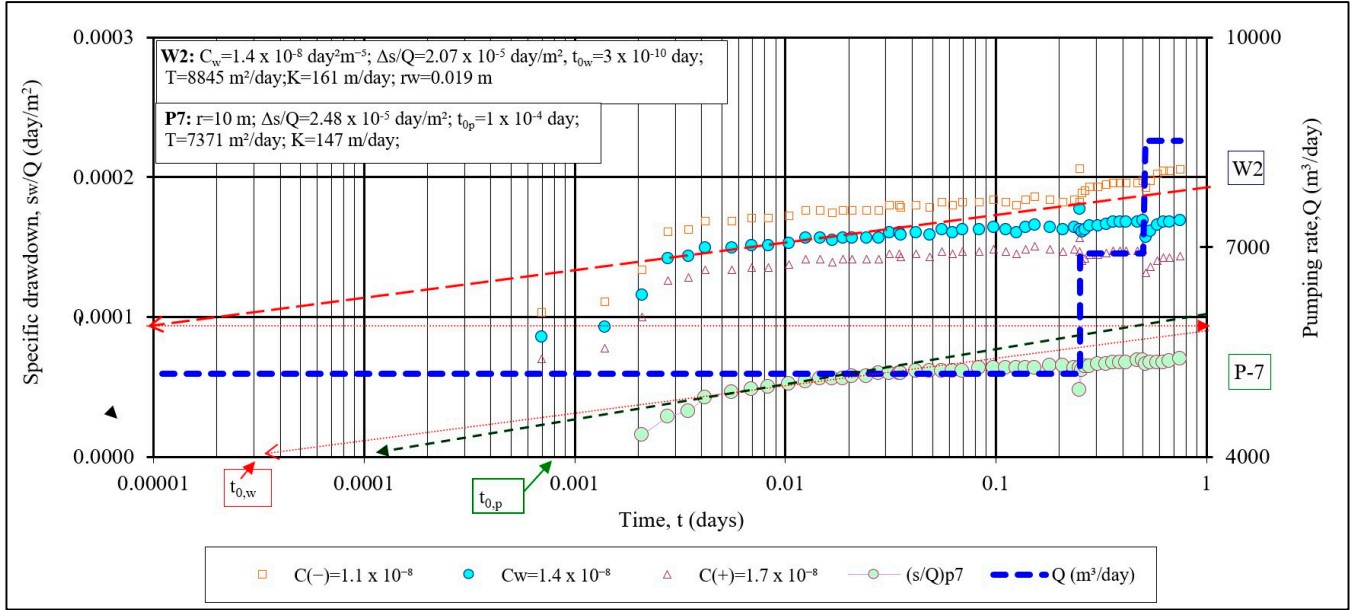

**Figure 5.** Specific drawdown $s/Q$ of linear losses during the step-drawdown test of the well W2 for real, underestimated and overestimated magnitudes of nonlinear losses C and piezometer P-7.

### 3.2. W2—Đurđevac

Pumped wells in unconfined aquifers are exposed to various boundary conditions, which, in addition to the occurrence of radial drainage of the aquifer, make it difficult to easily perform typical identifications of the aquifer parameters.

Well ZĐ2 on the W2-Đurđevac test field is one example of well testing in an unconfined gravel aquifer, under the limited impact of pumping. The test field of the well and the satellite piezometer is located in the zone of regional reduction of the aquifer's longitudinal slope and is surrounded by channels connected to the meanders of the Drava River. Well screens were installed at four intervals with a total length of $l = 35$ m (Table 1), which is 66% of the affected aquifer. The screens were positioned in a way to miss the gravelly aquifer intervals with an increased content of finer fractions. The highest screen was positioned 7 m below the initial water table, causing the effect of partial penetration. During the conducted testing, the largest drawdown in the well was $s_{max} = 2.55$ m, so the relationship between the maximum drawdown and the initial saturated thickness of the aquifer was $s_{max}/h_0 = 0.0425$, and in relation to the interval of partial penetration, $s_{max}/l = 0.073$. For such a circumstance, it is acceptable to use the corrected drawdown of Equation (9) for the application of analyses using methods of confined aquifers. The process of identifying the parameters of the aquifer and well W2 was carried out following Jacob's [9] recommendations, which are more demanding than the USDI [17] standard.

During the first two minutes of testing ($Q = 60$ L/s), no changes in the water level were observed in the observation piezometer P7 (Figure 6), 10 m from the well. In the first three minutes of testing the W2 well, the development of the specific drawdown can be compared to the final effect of delayed yield. After the fourth minute of pumping, the specific drawdown in the well joined the straight sequence in the semi-log diagram $s_w/Q = f(log10t)$, manifesting the development of the specific drawdown as corresponding to a confined aquifer. During the first thirty minutes (0.02 days), a typical linear sequence of curves was traced, manifesting the effect of transmissivity of the aquifer, which could be representative for the identification of parameters (Figure 6). During the later continuation of testing, attempts to maintain the given capacity of the first step of testing were visible. After two hours of pumping, the effect became limited.

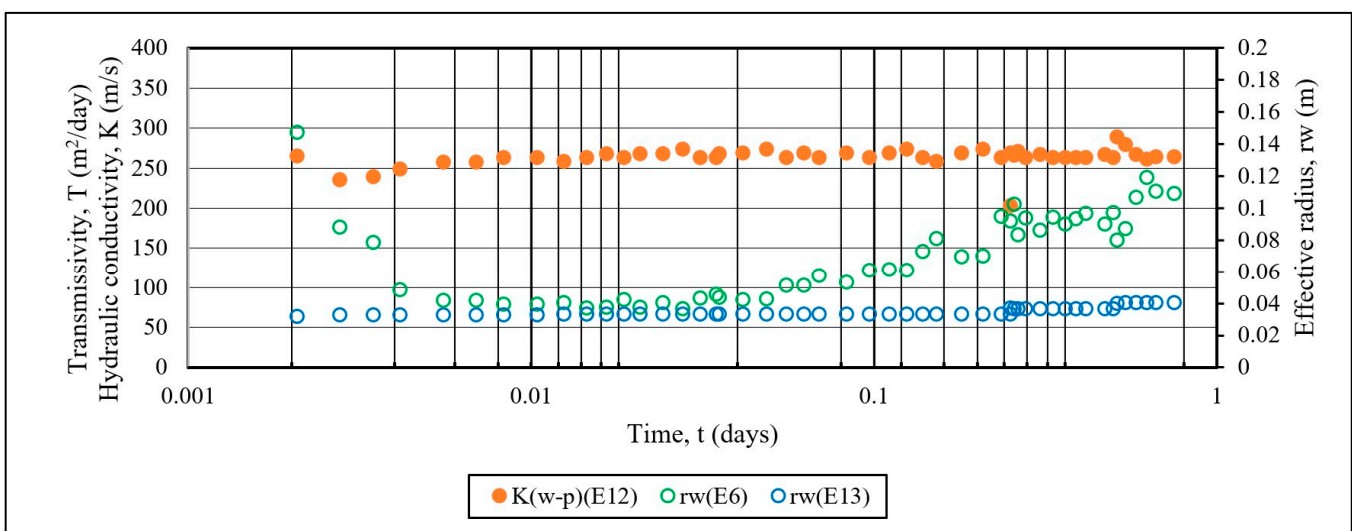

**Figure 6.** Hydraulic conductivity between pumped well W2 and observation piezometer P7 according to formula Equation (12), and effective well radius rw of W2 calculated from transient (Equation (6)) and steady state methods (Equation (13)).

When identifying the parameters, the corrected drawdown data (Equation (9)) were used. The identification procedure and the results of the parameters achieved by transient state methods are shown in Figure 6. The aquifer transmissivity coefficient calculated from

the graphical presentation of the transient flow, in Figure 6, was verified by analyzing the sequence of quasi-steady positions of specific drawdown between well W2 and piezometer P7, expressed by Equations (8) and (14) (Figure 7). The same figure shows the hydraulic conductivity results (Equations (12) and (15)), of which $K_{w-p}$ is close to the conductivity of the pumped well W2 in Figure 6, and the hydraulic conductivity of the initial aquifer $K_{h0}$ is similar to the conductivity result determined for P-1 by the transient method for P7 (Figure 7).

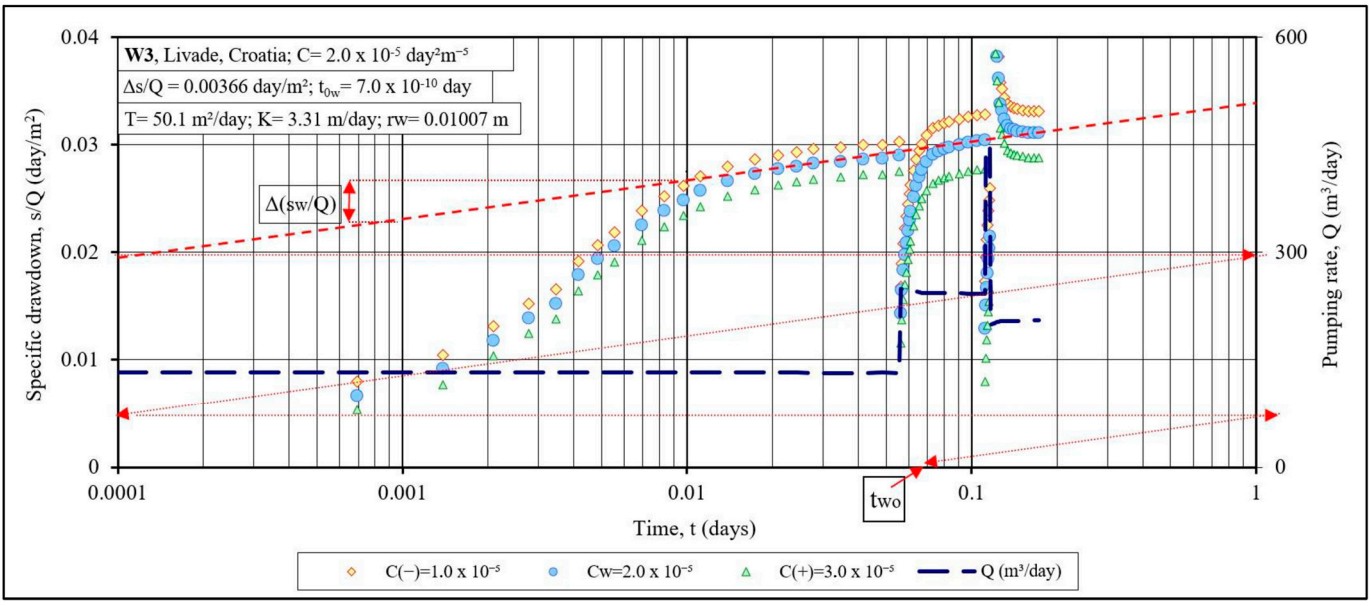

**Figure 7.** Continuity of specific drawdown on the example of the four-hour step-test of well W3.

### 3.3. W3—Livade, Beli Manastir

The third example, the interpretation of the pumping test of the W3 well (Figure 7), demonstrates the development of spontaneous specific drawdown without maintaining the constant yield of a single well. The well captures a confined sandy aquifer (Table 1) in the area of the Danube terrace (Figure 1). The pumping test of the well was carried out after its revitalization. The step-test of the well was performed with the means to avoid the problems faced when reinterpreting the data of earlier tests, by insisting on the demonstration of the effect of the spontaneous specific drawdown of W3 (Figure 7) with frequent measurements of the drawdown and capacity of the well. Under such conditions (Equation (3); Figure 7), testing provided a realistic presentation of the results of the impact of aquifer parameters as a function of specific drawdown as an objective function of aquifer properties.

The duration of the pumping test was shortened, and all three test steps were performed in a shorter period than the first step of testing wells W1 and W2. During each step of pumping of W3, the spontaneous development of $s$ and $Q$ was allowed, with consistent and frequent measurements. At the end of the second step, a short, strong disruption of the yield of the well (Figure 7) was performed, and then the pumping rate of the third step of the test was significantly reduced. Harmonized measurements of drawdown and the pumping rate of the well enabled the clarity of the presentation of the specific drawdown as a continuous function of linear losses independent of the pumping rate of the well.

All three testing steps offered a unique line of aquifer transmissivity and a unique coefficient of nonlinear losses of the well $C_w$. The results of the coefficient of nonlinear losses, $C_w$, transmissivity, $T$, and hydraulic conductivity, $K$, of the aquifer were realistic and verified (Figure 7), and the effective radius of the well, $r_w \approx (0.9 \div 2) * 10^{-3}$ (mm), was calculated using the storage coefficient estimated using the literature values of the compressibility coefficient of water and deposits at location W3.

### 3.4. W4—Daruvar Spa

Test field W4 was selected for the analyses described due its different features from the previously describes test fields, W1, W2 and W3. The aquifer captured was confined and secondary porous, and the well diameter was smaller than that previously described.

In 2009, the opportunity occurred to construct a new well of the fractured aquifer at the source of the thermal water in Daruvar spa in the western part of Mt. Papuk. The aquifer is built of calcareous (dolomite) breccias, sandstones and shales from 59 to 190 m deep. They are interspersed with fissures and caverns partially filled with calcite. Calcite and pyrite crystals also appear in the caverns. Above the dolomitic carbonates of the Upper Triassic, ten meters of compact Baden breccias follow, ending with lithotamnian limestones covered with thirty meters of thick Baden marls. The final Quaternary deposits are predominantly clayey-sandy in content, with occasional rubble and pebble content. The most favorable permeable intervals, between 128 and 134 and between 140 and 187 m deep, were screened. Slotted screens were installed through them (slots 3–4 mm wide and about 10 cm long).

The continuous testing of well W4 in steps lasted 180 min (0.257 day, Figure 8). The maximum pumping rate was 13.45 L/s, and the highest drawdown of the water level was 0.59 m. The technical circumstances at the pumping station made it difficult to conduct the standard step test with an increasing pumping rate. Nevertheless, by frequent measurements of the pumping rate of the well and the resulting drawdown of the water level in it, a realistic picture of the development of the specific drawdown was identified for the reliable identification of the coefficient of nonlinear losses and the transmissivity of the investigated aquifer (Figure 8).

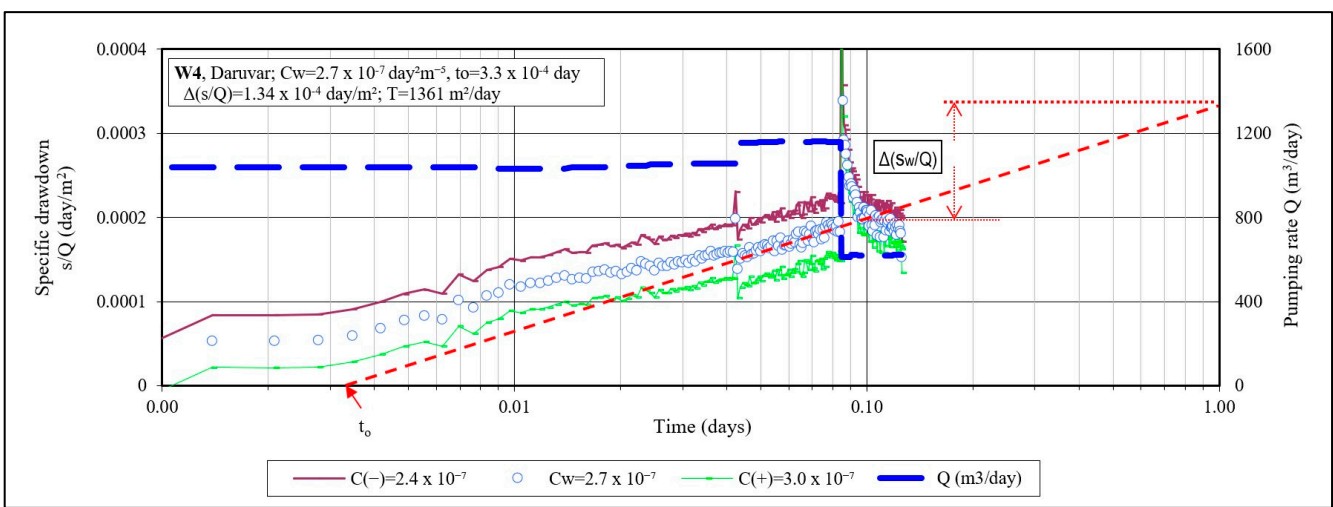

**Figure 8.** Specific drawdown during the step test of the W4 well.

### 4. Discussion and Conclusions

The formulation of the radial flow in the form of a specific drawdown brings special approaches in the implementation and analysis of the step-test and provides significant advantages in the identification of well and aquifer parameters. The basic rule of application of specific drawdown is the spontaneous development of the $s_w/Q$ ratio, the specific drawdown of linear losses in the well, as an integral parameter of the aquifer properties during radial flow, i.e., dependent only on the properties of the affected aquifer. Such conditions are controlled only by frequent measurements of yield $Q_i$ and measured drawdown $s_{m,w}$ (Equation (2)). In contrast to the models of drawdown of the radial flow while maintaining a constant pumping rate of the well, for the model of specific drawdown the use of spontaneous development of the capacity and the corresponding adjustment of the drawdown in the well was insisted upon. In order to achieve the spontaneous sequence of drawdown and pumping rate, it is a necessary to perform frequent and coordinated measurements of the capacity and drawdown in the pumped well. A rapid change in pumping rate causes a

discontinuity of the specific drawdown curve, revealing the magnitude of the coefficient of nonlinear losses of the well, $C_w$. The value of the coefficient $C_w$ regulated the balance of the distribution of linear and non-linear losses within the total measured losses, and thereby verified the actual value of the linear losses. This magnitude of the specific drawdown of linear losses below the initial level in the pumped well corresponded to the position of the effective radius, which was also the position of the respective list formed of linear losses of the specific drawdown $s_{w,t}/Q_t = B_{rw,t}$. This rearranged the water level measurement data in the pumped well into a data sheet. The lowest level at the position $r_{w,t}$ was the maximum drawdown in the spatial development of the radial flow for each moment of the step-test measurement.

The specific drawdown in the radial flow is a joint function of two mutually inversely proportional variables: the pumping rate of the well and the resulting drawdown, $s_{w,t}/Q_t$. Their ratio gradually changed during aquifer depletion, and this change depended only on the aquifer's properties. The magnitude of the function $s_{w,t}/Q_t$ was independent of the well capacity. The rapid change in the pumping rate of the well demonstrated the inertia of the drawdown adjustment to the new pumping rate appropriate to the aquifer outflow, marking a continuous sequence towards the development of the specific drawdown of the linear flow in the previous step of the step-test. The $s_{w,t}/Q_t$ ratio spontaneously adapted to the small and gradual changes in the pumping rate. Before and after each abrupt change in pumping rate, positive as well as negative, the path of specific drawdown continued unchanged, identical to the continuous path of specific drawdown (Figures 2, 5, 7 and 8) governed by the properties of the aquifer. Only such a reaction unambiguously reveals the exact size of the coefficient of non-linear losses $C_w [T^2 L^{-5}]$. An underestimated value of $C-$ indicated a too high specific drawdown, and an overestimated value of C+ caused too low specific drawdown of the linear flow of the next step of the step-test. The semi-log diagrams of these two curves (Figures 2, 5, 7 and 8) formed a corridor through which the path of real specific drawdown passed, for which in Equation (2) $C = C_w$, as well as the equation of the specific drawdown of the linear radial flow of the investigated aquifer. In this case, Cooper and Jacob's [1] equation was used, in which the drawdown function was rearranged into a specific drawdown function (Equations (2) and (3)).

The identification of aquifer parameters from test well pumping data by the analysis of the specific drawdown of radial flow enabled the tracking of the effect of all the parameters on the development of the cone of depression regardless of the well pumping rate, except for immediately after a rapid change in yield. Specifically, this rapid change in productivity was subject to the strong effect of the inertia of the tested system when transforming the amount of specific drawdown from the previous step to the same amount, $s/Q$, in the next step, thus enabling the identification and verification of the coefficient $C_w$ of the tested well. With the continuation of pumping, the specific drawdown continued to spontaneously tend to the same value, characteristic of the tested aquifer, continuing to express the real effect of its parameters. These are the circumstances that generate the special importance of systematic measurements of the pumping rate of the well and the appropriate sequence of measuring the water level in it. By applying the analysis of specific drawdown, the pumped well becomes the versatile backbone of the observation system of the tested aquifer. Therefore, it is imperative to adapt the water level measurement frequency to dense measurements in the first period of every pumping rate.

The unequivocal determination of the coefficient of non-linear losses through the specific drawdown model and the graphical verification of the results are an elaborative confirmation of the formation of a real function of the development of linear flow through the aquifer cleared of the effect of non-linear losses in the pumped well. The spontaneous development of such a flow through the aquifer enables the identification of the real values of the aquifer parameters.

**Supplementary Materials:** The following supporting information can be downloaded at: https://www.mdpi.com/article/10.3390/su151411170/s1, Supplementary_Material, sheets: W1, W2, W3, W4.

**Author Contributions:** Conceptualization, K.U. and J.T.; methodology, K.U.; investigation, K.U.; data curation, K.U. and I.K.; writing—original draft preparation, K.U.; writing—review and editing, J.K.; visualization, I.K. and J.K.; supervision, K.U. and J.T. All authors have read and agreed to the published version of the manuscript.

**Funding:** This research was funded by the Croatian Science Foundation (HyTheC project; grant number UIP-2019-04-1218).

**Data Availability Statement:** Data available on request from the authors.

**Acknowledgments:** The authors would like to thank Kosta Urumović sr. and Marco Pola for fruitful discussions and consultation during the conceptualization of the research, and Hrvoje Čuljak for support during the research. Also, the authors are grateful to the Croatian Geological Survey for the logistic backup during the research.

**Conflicts of Interest:** The authors declare no conflict of interest.

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
