# Peer review of "Identification of Aquifer and Pumped Well Parameters Using the Data Hidden in Non-Linear Losses"

_sustainability, doi:10.3390/su151411170_

Round 1

Reviewer 1 Report

It is a very interesting research about the identification of aquifer and pumped well parameters using the data hidden in non-linear losses. 

I have just one question about the anayzed test fields. Why did they selected the stated area in Figure 1, especially?

Author Response

Rev 1:

I have just one question about the analyzed test fields. Why did they selected the stated area in Figure 1, especially?

Authors:

Locations were selected because they represent different hydrogeological conditions and were investigated through different testing approaches, respectively:

W1 – confined deep sandy aquifer; usual pumping test with sporadic pumping rate measurement;

W2 – unconfined gravelly aquifer usual pumping test with sporadic pumping rate measurement;

W3 – confined shallow sandy aquifer; pumping test with frequent pumping rate measurement;

W4 – confined fractured karst aquifer with frequent pumping rate measurement;

These hydrogeological settings are most commonly investigated and testing approaches are the most used in hydrogeological works. Furthermore, all the described wells were either built and tested (W1, W2, W3) or just tested (W4) under authors’ supervision.

Reviewer 2 Report

Manuscript ID: Sustainability-2479981

In the current exploration, the authors investigate Identification of aquifer and pumped well parameters using the data hidden in non-linear losses. The work seems to be interesting, original, and also well-developed. Figures are also included for better improvisation. I recommend the publication of the manuscript after minor changes. My suggestions are listed below:

Ø  What is the main contribution addressed by authors regarding the field?

Ø  Abstract should be enhanced with major results.

Ø  References are appropriate but it will be better if authors add some recent papers regarding to your study. The authors have added very old papers.

Ø  Add few lines about methodology regarding the advantages.

Obtained results should be shown in Tables if possible.

Minor editing of English language required

Author Response

Rev 2:

  1. What is the main contribution addressed by authors regarding the field?

Authors:

The main contribution regarding the field is a novel approach to step drawdown tests interpretation, resulting with a substantial increase of precision of non-linear well loss coefficient determination. Also, the emphasis on the reviling of the pumped well as the main observation object for the purpose of aquifer properties determination enables simpler and more costly efficient identification of hydrological conditions in hydrogeological researches. 

these sentences are added in the "Introduction" (L. 52-57)

  1. Abstract should be enhanced with major results.

Authors:
We have slightly shortened the introduction and added one more advantage of the method but we don’t believe that results should be presented in the Abstract.

  1. References are appropriate but it will be better if authors add some recent papers regarding to your study. The authors have added very old papers.

 Authors:

The interpretation of the step-drawdown data has been developed long time ago, and experts and scientists have always cited these original works. There are a few rather recent methodological papers on the topic, but they employ a different approach than our work. Nevertheless, we have added two in the appropriate place of the text (L. 97 in the revised version of the manuscript).

  1. Add few lines about methodology regarding the advantages

Authors:

A few lines regarding the advantages of described methodology were added in the 2nd chapter (L. 257-260).

Reviewer 3 Report

Dear authors, I have already checked your manuscript entitled "Identification of aquifer and pumped well parameters using the data hidden in non-linear losses". After reviewing the manuscript. I found that the main contributions and benefits of your research. The recommendations for this decision are as follows:

1. In line 21, “…. the aquifer parameters relations….” should be changed into “…. the aquifer parameter relations….”.

2. In line 43, “. As opposed to to linear…” should be changed into “As opposed to linear…”.

3. In line 47, “2. Description of the test fields and analyses methodology” should be changed into “2. Description of the test fields and analysis methodology”.

4. In line 58, “…building of the respective wells in 2010. and 2013.” should be changed into “…building of the respective wells in 2010 and 2013.”.

5. The author needs to explain in detail in the "Introduction" what is the effects of aquifer and pumped well parameters. Moreover, the significance and progress of your research needs to be explained in "introduction" section. Some references should be cited as follows: (a) Synergistic effects of dodecane-castor oil acid mixture on the flotation responses of low-rank coal: A combined simulation and experimental study. International Journal of Mining Science and Technology 2023; 33:649-658. (b) Investigation of collector mixtures on the flotation dynamics of low-rank coal[J]. Fuel, 2022,327: 125171.

6. The “Conclusions” and “Introduction” sections should be refined and shorten.

7. The quality of the figures should be improved.

8. Authors should carefully check the format of references and citations.

no.

Author Response

Dear authors, I have already checked your manuscript entitled "Identification of aquifer and pumped well parameters using the data hidden in non-linear losses". After reviewing the manuscript. I found that the main contributions and benefits of your research. The recommendations for this decision are as follows:

  1. In line 21, “…. the aquifer parameters relations….” should be changed into “…. the aquifer parameter relations….”.

The authors agree, changes made as requested

 In line 43, “. As opposed to to linear…” should be changed into “As opposed to linear…”.

The authors agree, changes made as requested

 In line 47, “2. Description of the test fields and analyses methodology” should be changed into “2. Description of the test fields and analysis methodology”.

The authors agree, changes made as requested

 In line 58, “…building of the respective wells in 2010. and 2013.” should be changed into “…building of the respective wells in 2010 and 2013.”.

The authors agree, changes made as requested

  1. The author needs to explain in detail in the "Introduction" what is the effects of aquifer and pumped well parameters.

The authors don’t understand the reviewers request

Moreover, the significance and progress of your research needs to be explained in "introduction" section.

One paragraph added in the Introduction chapter

 Some references should be cited as follows: (a) Synergistic effects of dodecane-castor oil acid mixture on the flotation responses of low-rank coal: A combined simulation and experimental study. International Journal of Mining Science and Technology 2023; 33:649-658. (b) Investigation of collector mixtures on the flotation dynamics of low-rank coal[J]. Fuel, 2022,327: 125171.

The authors agree, changes made as requested

  1. The “Conclusions” and “Introduction” sections should be refined and shorten.

The “Introduction” chapter is rather short and we don’t see a need to shorten it.

The “Conclusions” chapter is rather long just because we have combined it with the Discussion chapter. We believe that there is no need to shorten it. However, if the Reviewer 3 thinks that it is too long, we could try to shorten it if the Reviewer provides more detailed comments on the parts that are not needed.

  1. The quality of the figures should be improved.

The authors agree, changes made as requested

  1. Authors should carefully check the format of references and citations

The authors agree, changes made as requested

Reviewer 4 Report

Just to clarify the suggestion for (minor) textual improvement, the first sentence presently reads:

"during the pumping of the well the groundwater level drawdown ... "

which could be better formulated in more general terms

"during the pumping of wells, the gw level drawdown .. "

Hence the suggestion to have the manuscript checked by a native (technical) English speaker.

see above

Author Response

Rev 4

ust to clarify the suggestion for (minor) textual improvement, the first sentence presently reads:

"during the pumping of the well the groundwater level drawdown ... "

which could be better formulated in more general terms

"during the pumping of wells, the gw level drawdown .. "

Hence the suggestion to have the manuscript checked by a native (technical) English speaker.

Authors:

The authors thank to Reviewer 4 for his comment, and suggestions. We have checked English and have made few minor changes. If the Editor thinks that we should send the manuscript to Language Edit, we will do as requested, but it will take time to get the edited text back.

Round 2

Reviewer 3 Report

The paper has been well modified and can be accepted.

no